# Participatory Design of Nature-Based Solutions: Usability of Tools for Water Professionals

**Borjana Bogatinoska** [1,*] , **Angelique Lansu** [1] , **Jean Hugé** [1] **and Stefan C. Dekker** [1,2]

1 Department of Environmental Sciences, Faculty of Science, Open University of the Netherlands, 6419 AT Heerlen, The Netherlands; angelique.lansu@ou.nl (A.L.); jean.huge@ou.nl (J.H.); s.c.dekker@uu.nl (S.C.D.)

2 Department of Environmental Sciences, Copernicus Institute, Utrecht University, 3584 CB Utrecht, The Netherlands

* Correspondence: borjana.bogatinoska@ou.nl; Tel.: +31-455-762-877

**Abstract:** Participatory processes provide opportunities for water professionals such as scientists and policymakers and other stakeholders such as the local communities and farmers to meet, exchange information, deliberate, and share values. There is a diversity of rapidly evolving participatory methods, here defined, as tools for supporting the process of designing nature-based solutions (NbS) together with the stakeholders. However, this requires a systematic and informed selection to facilitate the adequate choice of tools, aligned to the requirements and context of the water professionals and the stakeholders for the design and deployment of NbS. Despite this, there is still little progress and knowledge accumulation on how to select the most context-appropriate tool(s). Consequently, in this research, we propose a stepwise framework for the use of participatory tools, which we categorize as: (i) tools used for defining the hydro-meteorological hazards (HMH) and its impact on stakeholders—knowledge tools (ii) tools used for co-designing NbS with stakeholders—co-creation tools and (iii) tools used for co-implementing the transition towards NbS—transition tools. We then apply and test this stepwise framework on the participatory processes used in eight brook catchments distributed in four countries: the Netherlands, Belgium, France, and the United Kingdom. The framework is designed in steps that would lead to respectively: selecting, classifying, mapping, and grading the participatory tools leading to an informed and systematic decision of a tool or suite of tools for the design and deployment of NbS with stakeholders. With the application of this framework, we see that among the water professionals: (1) knowledge tools are central in the problem definition stage, particularly with non-technical stakeholders; (2) most anticipated co-creation tools are e-Tools/Virtual tools and workshops; (3) transition tools favor visual tools as a way of enabling the transition towards management practices.

**Keywords:** hydrometeorological hazards; co-design; stakeholders; north-western Europe; decision support; usability index; brook catchment

## 1. Introduction

In the north-western part of Europe, global warming and the resulting intensification of the water cycle have been associated with the increase in frequency and magnitude of extreme hydrometeorological hazards (HMHs), by more extreme heavy rainfall and a greater chance of prolonged drought or heat [1,2]. In most cases, HMHs are caused by a combination of naturally occurring extreme weather events and anthropogenic activities [3,4].

Our focus is on brook catchments in the north-western part of Europe. Brook catchments are local ecosystems that are part of larger regional ecosystems. These brook catchments are the upper tributaries of lowland rainfed rivers like Meuse and Scheldt. Other than these meandering rivers with their nearly flat floodplains, brook catchments have narrow valleys, making them more vulnerable to being affected by HMH [5]. By extreme summer storms causing flash floods over land, brook catchments react with high peak flows in a

short time with risks of flooding, instead of spreading peak flows over a longer period as rivers with floodplains do. The shorter retention time also makes these catchments more vulnerable to drought [6].

Catchments can be regarded as a socio-ecological system (SES) as these include stakeholders which are affected in various ways by HMHs. In turn, the stakeholders' activities affect the catchment and its resilience to HMHs. An SES is a system of biophysical and social factors that regularly interact in a resilient, sustained manner [7–9]. Vulnerable catchment-SES have lost their resilience and with it their adaptability towards changing external drivers like extreme climate events [8].

Nature-based solutions (NbS) are increasingly being implemented as suitable approaches for reducing vulnerability and risk of SES to HMHs [4] in contrast to man-made 'hard' engineering solutions. IUCN (2017) [10] has defined Nature-based solutions (NbS) as actions to protect, sustainably manage and restore natural or modified ecosystems, which address societal challenges such as climate change, food, and water security, or natural disasters effectively and adaptively, while simultaneously providing human well-being and biodiversity benefits. NbS, despite gaining vast popularity in contrast to 'hard' engineering solutions, is still an emerging concept that requires more evidence-based results on their long-term effectiveness [11].

The full impact of global change and the expected increase of HMH on catchments in northwestern Europe remain uncertain [12,13]. Therefore, any NbS should be able to change over time as well—become adaptive. In contrast with many 'hard' engineering solutions, the 'soft' nature-based solutions have the potential to deal with both climate mitigation and adaptation challenges at a relatively low-cost while delivering multiple additional benefits for people and nature [14].

When a planning process, in our case that is the planning and designing of NbS, includes stakeholders, it is considered a participatory process [15]. Participatory processes provide opportunities for scientists, policymakers, stakeholders, and citizens to meet, exchange information, deliberate, and share values [15–18]. The use of the participatory design of NbS has been considered a good practice among researchers [19–21]. The artifacts through which the water professionals (scientists and policymakers) and the other stakeholders such as citizens can enable these participatory processes are defined as tools. These tools often provide the necessary information and a common language through which issues can be discussed without constraining creativity and the open explorations of ideas [16]. As a result, many open-source or licensed tools databases have been developed to guide the implementation of climate change adaptation measures such as NbS [15,17,22].

Despite the potential of tools to support the participatory process, there are still knowledge gaps and barriers that are hindering their contribution to NbS projects at a local level by cities and local authorities [15]. In their review of participation tools, van Asselt and Rijkens-Klomp (2002) [23] stated that not enough systematic attention was being paid to the selection and implementation of participation tools and that more effort was required to explicitly state the tools and lessons learned from participatory processes. McEvoy et al. (2018) [15] conclude that there is a need for in-situ studies, and question whether tools can be interchangeable or not in a participatory process. Moreover, Salter et al. (2010) [16] stated that there is fragmentation over preferred tools which are often ad hoc, prompt exercises where much of the long-term value is lost by the cessation of projects and the necessity of the next funding cycles.

There are still wide hurdles when it comes to the participatory design of NbS specifically in brook catchments as opposed to urban areas. Researchers call for a better grasp of individual tools having considered their effectiveness and the conditions under which they are suitably used [17,24–26]. Voskamp et al. (2021) [22] look into tooling used for climate adaptation with NbS but merely for planning in urban areas. They suggest a user-oriented framework for tool selection by looking into the sustainability aspects and NbS benefits from each specific tool.

From the above review of current practices, we conclude that it is important to consider the characteristics of the participatory tools and their suitability for specific cases and conditions. It also raises the question of whether these tools are giving added value to the participation processes or not. Water professionals and the other stakeholders can only benefit from these tools when they are: (i) cognizant of their existence, (ii) can compare the varied available tools, (iii) can make an informed selection of the tools suitable to reach specific objectives and stages in the process in their catchments and (iv) adapt them to their specific needs and local contexts. Furthermore, there is also more information needed on how water professionals can facilitate these tools.

For this purpose, we propose a stepwise framework to categorize different tools in the participation processes in a systematic way and develop a methodology for calculating the usability index per tool. Furthermore, we test such a stepwise approach in order to analyze the participation processes in eight catchments in north-western Europe, particularly in France, the Netherlands, Belgium, and the United Kingdom. This will result in a showcase of the lessons learned from using the suggested framework and participatory tools. While insightful reviews of tooling used in climate adaptation exist [22], this paper focuses on a user-oriented framework for participatory tools used not only in urban but also in rural surroundings. Hence the significance lies in two reasons. As far as we are aware, it is the first to propose a framework for making an informed and systematic selection of participatory tools for the design of NbS and establish a set of principles and lessons learned for the use of participatory tools in brook catchments that could be applied for participatory design in different sustainability contexts. By doing this, we will answer the following research question:

How can water professionals make an informed decision on which tools to use when designing NbS with stakeholders?

## 2. Study Area: Eight Brook Catchments

### 2.1. Socio-Ecological System of Brook Catchments

Local management practices, such as NbS, which shape the catchments, are formed by institutions and regulations on a regional, national, and European level. These management practices are operated by water professionals who are working with other stakeholders in the catchments [27]. Stakeholders in our research are defined as individuals, groups, and organizations who are affected by or can affect the design, implementation, and evaluation of measures in their catchment [28]. To incorporate stakeholders and their activities into management practices, often participatory processes are used to define HMH problems and to co-design interventions [27]. Participatory processes collectively engage all stakeholders in the process towards a more resilient catchment-SES [27]. In our case, we refer to water professionals as stakeholders that are responsible for the facilitation of the participatory processes. In this research, we will call them water professionals, to separate them from the rest of the stakeholders in the area (Figure 1). The water professionals (WP) and the rest of the stakeholders' co-design the NbS via participatory processes.

Further, we describe the brook catchment system as a Socio-Ecological-System (SES) and connect the natural ecosystem processes with the social processes. Conceptual visualization of how these SES are constituted is presented in Figure 1. For every catchment, we have 2 subsystems that constitute the SES. The ecosystem is presented with a green color. In our case, this is the brook catchment that is initially affected by HMH, namely flooding, drought, or waterlogging (i.e., saturation of soil with water). To achieve a transition towards a healthy and resilient catchment, different types of NbS can be implemented. This is where the social systems interfere. WPs are practitioners coming mostly from government or NGO organizations that have the power to select and use tools in the process of participatory design of NbS. The interaction between these 2 subsystems; the ecosystem and the social system, results in: (i) multiple ecosystem services provided from the ecosystem to the social system, and (ii) interdisciplinarity in terms of the design of the NbS where stakeholders contribute their knowledge and skills to manage the ecosystem—which is our area of

interest in this research. Bellow, we will shortly explain the eight different catchments in the context of their SES.

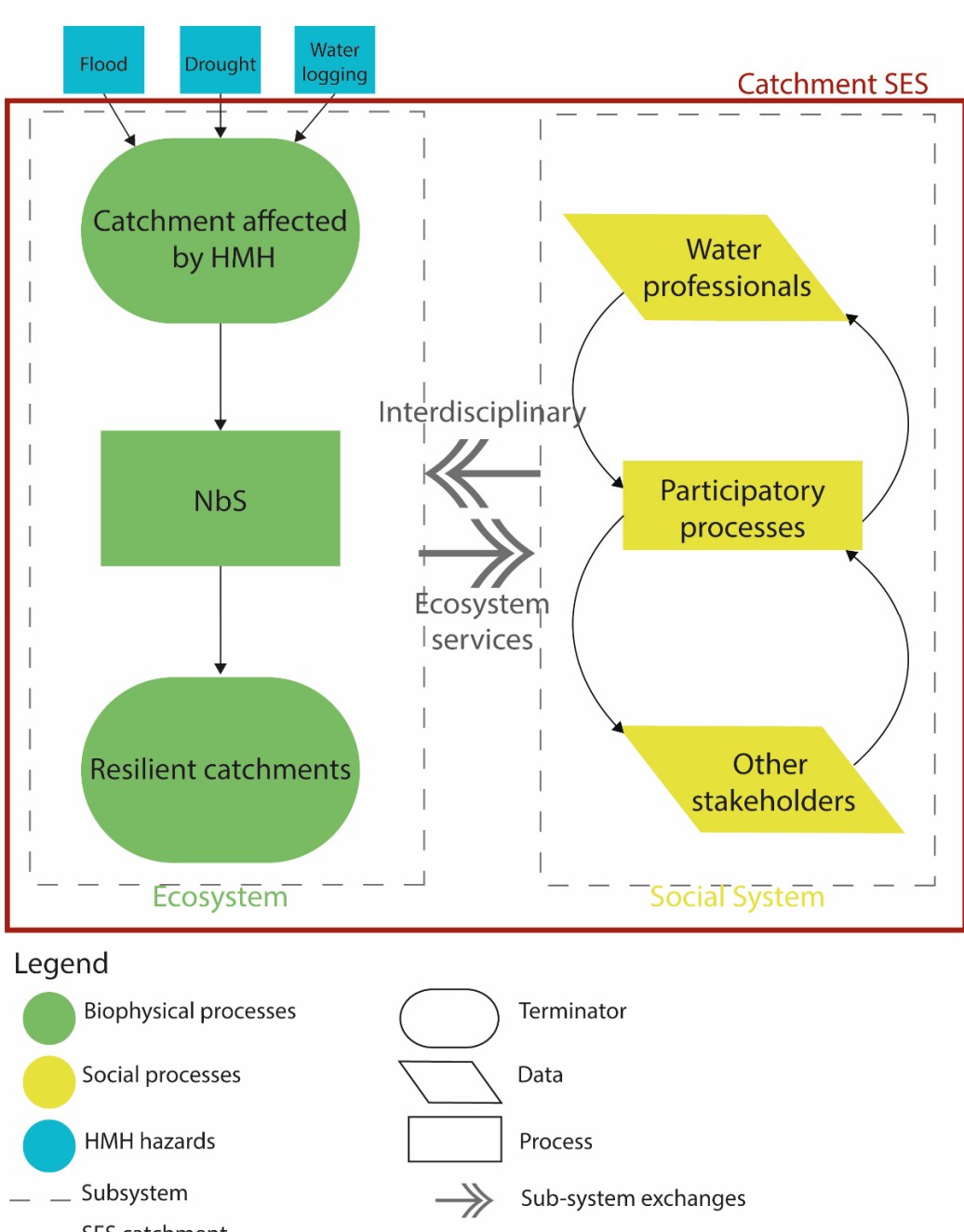

**Figure 1.** Conceptual understanding of the brook catchments and its stakeholders as a socio-ecological system.

### 2.2. Site Selections

The participatory design of the NbS framework suggested in this research is implemented and tested on the tooling used in a bigger Interreg 2 Seas project, the Co-Adapt: Climate adaptation through co-creation project. The overall objective of Co-Adapt is to develop, test, and roll out approaches to co-design of NbS to improve the adaptive capacity of the 2 Seas region to the water-related effects of climate change (Figure 2 and Table 1).

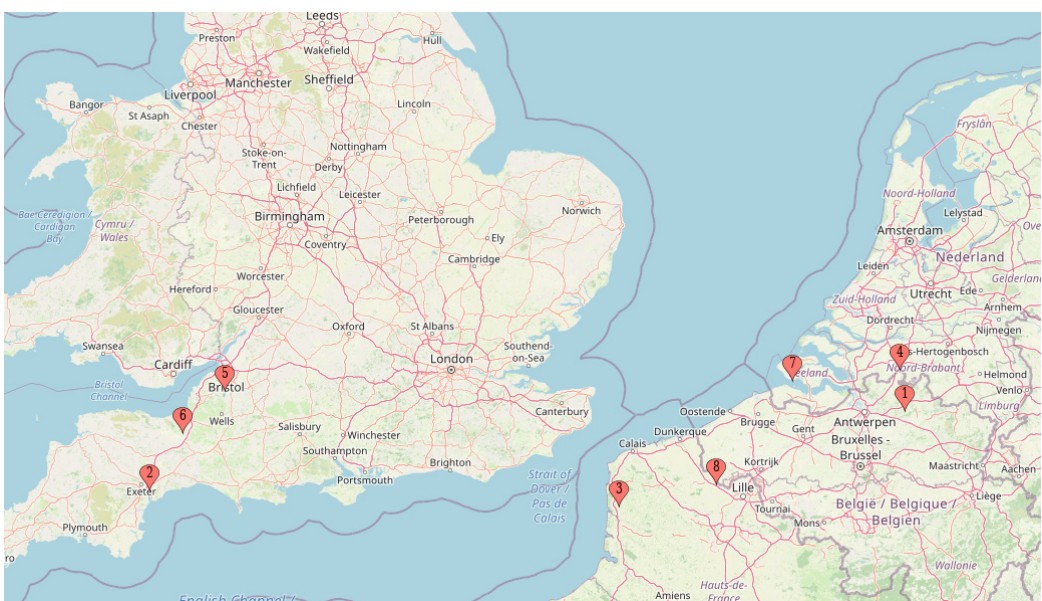

**Figure 2.** The Co-Adapt project partners and their pilot catchments (numbers correspond to the characteristics given in Table 1).

**Table 1.** Selected catchments and some key characteristics.

| No. | Catchments | Country | Key Characteristics |
|---|---|---|---|
| 1 | Laakbeek | Belgium | Small brook passes through the semi-urban village with flooding history. Not much space for adaptation measures. |
| 2 | The Culm | The United Kingdom | Brook passes a new 'green' development area and main railway. Flooding is the main HMH causing deterioration in water quality. |
| 3 | Liane | France | Brook passes a rural area with urbanized banks causing flooding and soil erosion. |
| 4 | Aa of Weerijs | The Netherlands | Brook passes a rural area with a high density of tree nurseries for export. Main HMH is drought due to high water demand and flooding in moments of peak flows. |
| 5 | Porlock Vale | The United Kingdom | Brook passes a steep valley, creating a high risk of flooding in several villages. |
| 6 | Somerset Levels and Moors | The United Kingdom | Flooding at lower reaches in several villages. |
| 7 | Vlissingen | The Netherlands | Channelized brooks pass through the new 'green' development area. Flooding is the main HMH. |
| 8 | West Flanders | Belgium | Brook passes between two villages and has a history of flooding. |

## 3. Materials and Methods

### 3.1. A Stepwise Framework for Tool Selection

For the purpose of developing a framework that would aid future WP in making an informed selection of participatory tools for their catchments, we developed a mixed-methods approach. By following them, WPs or other practitioners would get an overview of the existing 'market' of tools and their characteristics, get a quantitative indicator on how easy or complex the tool is with the usability index and decide whether they will use the suggested tools in the NbS handbook or develop new tools based on these findings.

In the first step, we designed questionnaires based on a literature review and we conducted eight semi-structured interviews with practitioners working in the area to collect all the tools used by the WPs. In Step 2, we organized webinars and workshops with the

WPs to clarify these questionnaires and discuss the nature of the tools and classify them into general groups based on their aim. In Step 3, we conducted additional semi-structured interviews with three practitioners to further enrich the data we already collected on tools and clear out some ambiguities from the questionnaires. Finally, in Step 4, these findings were synthesized and mapped in a tool framework that aims to provide potential end-users not only with an overview of existing tools but also indicates how these tools can support them to address specific challenges in their projects based on a set of criteria obtained with a literature review. In the following sections, these steps are described in detail.

### 3.2. Step 1: Tools Collection and Selection

NbS participatory tools were identified through: (i) questionnaire (*n* = 24), (ii) additional semi-structured interviews conducted with a few water professionals from the Co-Adapt project (*n* = 8), and (iii) additional desk research, as further detailed below.

The desk study entailed reviewing the websites and literature provided in the questionnaires by the WPs on the respective tools. This was done to fill in the gaps in the questionnaires and to provide additional information for the calculation of the usability index explained in the results section.

The questionnaire was designed with both fixed multiple-choice questions (MC) and open questions (OQ). Whereas primarily qualitative descriptions were used for all the tool characteristics, we still tried to use fixed indicators for the multiple-choice answers for easier further analysis and comparison. In Table 2, an overview of the questions and the possible answers (for the MC questions) are presented.

**Table 2.** Criteria used for structuring the questionnaire (MC—multiple-choice and OQ—open questions).

| Question | | Type | Criteria |
|---|---|---|---|
| Catchments | | MC | - |
| Communication type | | MC | Ad hoc, One-way, Two-way, stakeholder engagement, participatory decision making [29,30] |
| Type of stakeholders | | MC | Non-technical stakeholders (farmers, landowners, local communities; Non-agricultural businesses); Technical stakeholders (Educational and research institutions; NGO's; Governments; Chamber of Agriculture) |
| Description of tool | | OQ | Used to classify tools as: Knowledge Tool (KTx), Transition Tool (TTx), Co-Creation Tool (CCTx) |
| Primary Objective | | MC | Improve system understanding; Identify indicators and criteria; Identify issues, preferences, management options; Communication of knowledge; Identify knowledge gaps; Obtain information from stakeholders; Creation of knowledge with stakeholder; Evaluation (adapted from [17]) |
| NbS Design Process Stages | | MC | Actor analysis; Motivation; Problem Definition; Project Definition; Action Plan; Implementation; Evaluation (adapted from [31] and [32]) |
| Other Stages | | MC | Actor analysis; Motivation; Problem Definition; Project Definition; Action Plan; Implementation; Evaluation (adapted from [31] and [32]) |
| Level of participation | | | /Informing, consulting, involving, collaborating, empowering (adapted from [33,34]) |
| Strengths (from practitioner's perspective) | | OQ | - |
| Weaknesses (from practitioner's perspective) | | OQ | - |
| Practical Considerations | Time | MC | Low (few days); Medium (few weeks); High (few months) |
| | Budget | MC | Low (<100 euros); Medium (<1000 euros); High (>1000 euros) |
| | No. participants | MC | Small (<20); Medium (<100); Big (>100) |

The questionnaire was designed by using MC and OQ based on a literature review and interviews with the WPs. The choice of using multiple-choice questions was for practical reasons namely Step 4 of the stepwise framework-mapping of the participatory tools. We chose this so that an easier overview and classification could be given to the WPs for making an informed selection by using the stepwise framework and/or the NbS handbook on participatory tools.

The first classification used in the questionnaire is in terms of stakeholder communication. Our objective is to distinguish between tools that just provide certain information on the NbS in an ad-hoc manner to the point where they take the stakeholders' needs and preferences into account through different mechanisms of learning in the participatory decision making [29,30].

The second classification of the type of stakeholders is based on their technical knowledge. In consultation with the WPs, we decided to make the distinction between technical and non-technical stakeholders for practical reasons in terms of the participatory tool selection needs.

The third MC question is the objective of the tool. A plethora of authors suggests a combination of options on 'participatory process objectives' [17,35,36]. These frameworks combined with the experience of the authors on participatory processes and the WPs from our project allowed the identification of ten main options.

The next question refers to the NbS design process. For this, we used a combination of the suggested six steps of planning NbS proposed by Albert et al. (2021) [32], the revised workflow of participatory modeling suggested by Voinov and Gaddis (2017) [31] and adapted it to a seven-step process. The idea is to combine the NbS planning framework with a more practitioner-oriented workflow for participatory modeling to adapt to the needs of our WPs in the project. By doing this, we suggested a seven-step design process of NbS namely: (i) actor analysis—all stakeholders are identified and invited; (ii) motivation—discussing the knowledge, data, and priorities of the stakeholders; (iii) problem definition—creating an understanding of the existing problems across spatial and temporal levels; (iv) project definition—creating visions and scenarios and assessing their potential impacts; (v) action plan—develop solution strategies of the design and implementation of NbS; (vi) implementation—implementation of adaptation measures involving NbS; (vii) evaluation—assessing the effectiveness of the implemented (or planned to be implemented) adaptation measures including their monitoring. We want to state that the process of planning and designing NbS measures is far from a linear process and could involve multiple iterations by the time the final design is concluded. Consequently, one tool could be used in more than one stage of the design process of NbS, that is why we included an extra question regarding in which additional stages could the tool be additionally used.

The level of participation issue can be traced back to Arnstein's (1969) "ladder of participation" [33]. For the purpose of our research, we have decided to use the 'The public participation spectrum' division. This framework was designed to be applied in diverse engagement processes and contexts [34]. The following levels of participation are suggested: (i) informing—providing the stakeholders with objective information that can assist in strengthening the public understanding of a problem; (ii) consulting—obtaining stakeholders' feedback on a predetermined decision or idea; (iii) involving—working directly with the stakeholders throughout the NbS design process to ensure public concerns are understood and considered; (iv) collaborating—actively partnering with the stakeholders in determining the problem and preferred solutions and (v) empowering—placing final decision-making in the control of the public [34,37].

The final classifications are the tools' practical requirements. These were unified by using fixed indicators such as low, medium, and high for the time and budgetary variables and small, medium, and big indicators for the number of participants.

After the completion of the questionnaires, we made a final tools selection based on the following criteria: (i) whether the tool is a tool based on the classification from Ferreira et al. (2020) [11] and (ii) whether the tool can be used in a different context. The

second criterion was used so that this stepwise framework could be potentially used in other contexts of tools selection for participatory design with stakeholders and not only the participatory design of NbS.

### 3.3. Step 2: Tools Classification

This step is considered separate from Step 1 because we want to stress the importance of identifying which tool falls into which category of participatory tools. These categories were developed in a webinar with all WPs from the different catchments. They initially discussed topics related to the questionnaires from Step 1 and how to fill them efficiently. Secondly, they discussed the different aims of the tools or suite of tools they used in their own catchment with the stakeholders' groups. After the webinar, the water professionals had the necessary guidance and information to classify the tools and fill in the questionnaires.

### 3.4. Step 3: Grading (of the Tools)

Our third step after all the tools are collected and classified is calculating the usability index of each tool. The inspiration for this index comes from the fact that different potential users of these tools—the WPs in our case, have different specific questions that require answers.

The vast range of stakeholders aimed to be involved in designing NbS projects includes a different selection of skillsets and backgrounds. Consequently, the tool usability indication is an important feature for users to consider in selecting suitable tools. Usability is an element of the broader field of user-experience design (UX) [38] and can be understood as a function of user goals and objectives in addition to the user's 'environment' [39]. For calculating this index, we use an existing framework from Dargin et al. (2019) [40] which was used for the comparable purpose of tool evaluation for water energy food nexus (WEF) assessment tools.

Criteria for calculating the usability index have been derived from existing tool comparison charts [40–44]. Table 3 is an overview of the scoring criteria and includes qualitative and quantitative definitions and justifications for each criterion.

**Table 3.** Rubric for calculating the usability index. Source: Adapted from [40].

| No. | Criteria | Score | Score Description | Justification |
|-----|----------|-------|-------------------|---------------|
| 1 | Open Access | 0 | Yes | Cost is an important consideration in defining the usability tools [45] hence the need for utilizing open access tools. |
| | | 1 | No | |
| 2 | Web Interface | 0 | Yes | Tools with web interfaces reach larger and broader audiences which would translate to increasing the potential to discuss more outcomes and enhanced understanding of menagement measures such as NbS [44,45] |
| | | 1 | No | |
| 3 | Data Granularity | 1 | High level; national level data | Data granularity refers to the extend of detali in a specific data point [40]. More granular data sllows more thorough system{s} modelling however it can be hard to find in open source environments [46]. |
| | | 2 | General data with sector specific information | |
| | | 3 | Localized sector data and localized technical data | |

**Table 3.** *Cont.*

| No. | Criteria | Score | Score Description | Justification |
|---|---|---|---|---|
| 4 | Data Accessibility | 1 | Data available for most developed and developing countries | Data acessibility is an ongoing challeng specifically for accurate design of NbS. Acessibility is always connected to the granularity – hardly accessible data points are more likely to be detailed and difficult to measure. |
| | | 2 | Data is hardly accessible for developing countries | |
| | | 3 | Difficult to access data, derivation might require modelling tools | |
| 5 | Number of Data Inputs | 1 | 0–15 | Data inputs entail how many inputs a tool requires for the design of NbS. |
| | | 2 | 16–32 | |
| | | 3 | 33+ | |
| 6 | Needed Subject Expertise | 1 | Expertise not needed | The expertise needed for the users to be able to actively participate/use the specific tool. |
| | | 2 | Needs an understanding of general subject matter | |
| | | 3 | Expertise and high skill needed | |
| 7 | Training Intensity | 1 | 1 day | Trainings come with additional costs, time and resources. They can include online tutorials, independent instructions, in-person trainings etc. |
| | | 2 | 2–3 days | |
| | | 3 | 1 week | |
| 8 | User-Defined Scenarios | 1 | Yes | User-defined scenarios give added value for the process of designing NbS however they add to the complexity of the tool. |
| | | 0 | No/N-A | |

### 3.5. Step 4: Mapping

This is the final and illustrative step in our stepwise framework. The intention is to allow WPs to visualize which tool could be used for their specific participatory process. We extended the visualization methodology used by Dargin et al. (2019) [40] and included graphs plotting 2 closely related characteristics of the tools on the vertical and horizontal axis as an added value the usability index is presented as a bubble graph with the radius showing the value of the usability index. The differing usability 'bubbles' aim to also indicate the varying levels of detail and depth the tool is able to provide [40].

## 4. Results and Discussion

Our results are synthesized in the following way: First, we suggest a process/framework for collecting and selecting (Step 1), classifying (Step 2), grading (Step 3), and mapping (Step 4) the plethora of existing tools used for participatory designing NbS in a stepwise approach. Then we test this stepwise framework using the Co-Adapt project as our area of research and give some general conclusions and lessons learned from the participation process in these eight catchments.

All steps explained in the methodology chapter and later suggested in the stepwise framework (Figure 3) have been implemented in the catchments of the Co-Adapt tooling. This resulted in an NbS handbook with all tools used in the Co-Adapt project, made in such a way that the WPs could use it systematically. By applying the stepwise framework, we have the following results:

### 4.1. Framework for Selection of Participatory Tools for Designing NbS

With the steps explained both in the methodology and results stage respectively, we can constitute the stepwise framework for the selection of the NbS tool.

Figure 3 is a flowchart representing the stepwise framework for the selection of NbS participatory tools. It begins with the collection and selection of the array of tools to be

filtered through the mapping process. The available tools are then categorized according to stakeholder type, tool type, the primary objective, and the stage in the NbS design process as primary categories used for the mapping and the communication type, level of participation, description, strength, and weaknesses as secondary categories provided in Table S2. Next, usability indexes are computed for each tool based on the eight criteria presented in Table 3 [40]. The last step is mapping. In the so-called 'bubble' scatterplot formations, tool categories, tool type, and tools' objective and the type of stakeholders, and the NbS design process act as the independent variables on the x-axis and y-axis respectively, while the usability indexes appear as dependent variables with the radius of the 'bubble' presenting the score itself.

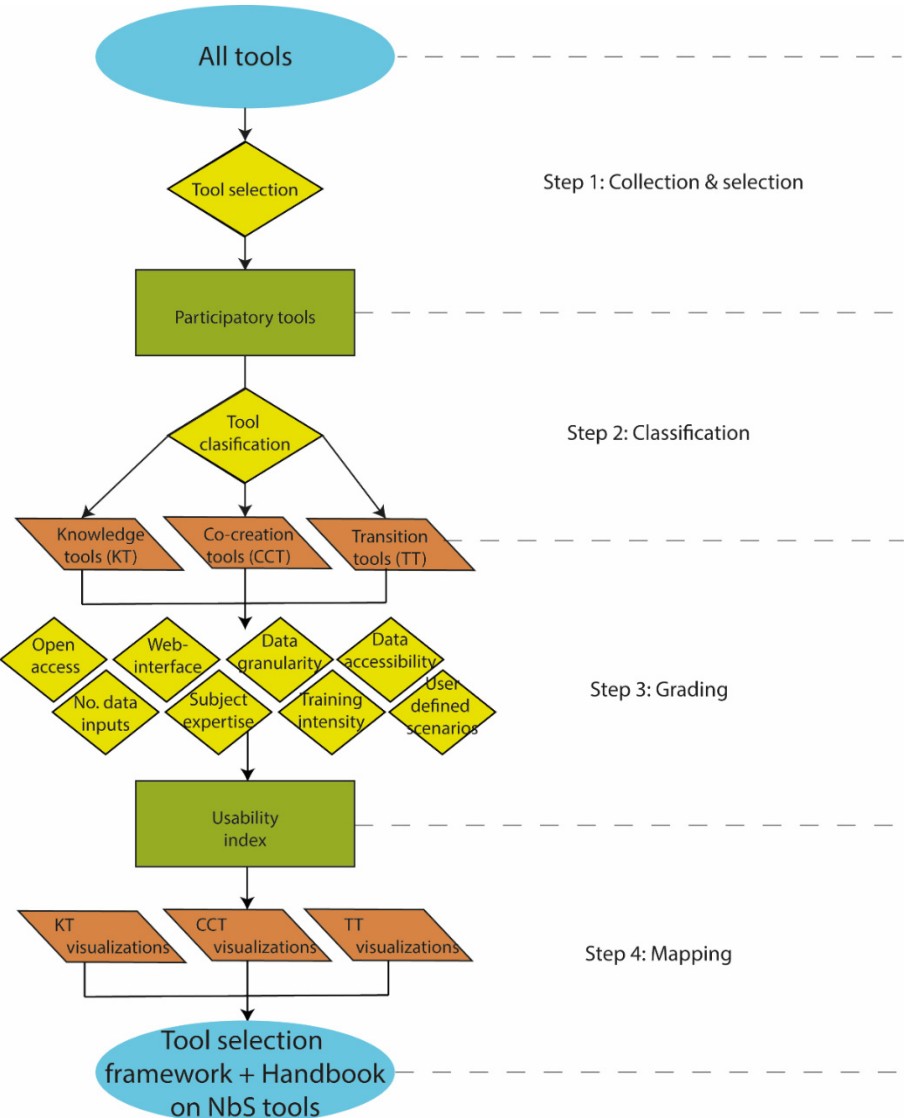

**Figure 3.** Stepwise framework for the selection of NbS participatory tools. Colors and shapes indicate the different functions, such as: blue/oval—start and endpoint of the framework; yellow/diamond—decision; green/rectangle—process and red/parallelogram—input/output.

### 4.2. Step 1: Collection and Selection

We started with collecting all tools used by the water professionals (Table S1: list of water professionals, on request in the Co-Adapt project). We gathered the information for each tool separately by using questionnaires (File S1) and then we conducted addi-

tional individual semi-structured interviews with the water professionals, for clearing out ambiguities from the questionnaires or filling in missing information where needed.

The selection stage consisted of two criteria: (i) whether the tool is a tool based on the classification from Ferreira et al. (2020) [11] and (ii) whether the tool can be used in a different context. Consecutively, from the total of 29 tools collected with the questionnaires, after the selection criteria, 2 tools were eliminated.

In Table 4 we give an overview of the selected tools and their codes/abbreviations. The codes are given based on their classification (KT—knowledge tools, TT—transition tools, and CCT—co-creation tools). One tool can be a part of more than one classification, hence the combined coding (example: KT9/CCT8).

**Table 4.** Overview of tools used in the study area catchments and their codes.

| Participatory Tool—title | KT | CCT | TT | Code |
|---|---|---|---|---|
| Hydrological model | ✓ | | | KT1 |
| Face to face field visits | ✓ | | | KT2 |
| 'Views on the climate issues' survey | ✓ | | | KT3 |
| Knowledge transfer website and newsletter | ✓ | | | KT4 |
| Public technical stakeholders' meetings | ✓ | | | KT5 |
| Public citizen meetings (Online) | ✓ | | | KT6 |
| Public citizen meetings (Physical) | ✓ | | | KT7 |
| Knowledge co-creation workshop | ✓ | | | KT8 |
| The walking app | ✓ | ✓ | | * KT9/CCT8 |
| Online ideation | ✓ | ✓ | | * KT11/CCT1 |
| Adaptation Pathways tool | ✓ | ✓ | ✓ | * CCT9/TT11/KT10 |
| Landscape planning | | ✓ | | CCT2 |
| Final design presentation | | ✓ | | CCT3 |
| Individual farm visits | | ✓ | | CCT4 |
| Digital collaboration tools | | ✓ | | CCT5 |
| The Forum | | ✓ | | CCT6 |
| Film Nights | | ✓ | | CCT7 |
| Flyer for planned future events | | | ✓ | TT1 |
| Stakeholder forum/Round table | | | ✓ | TT2 |
| Permanent information plaques and project area accessibility | | | ✓ | TT3 |
| Educational trainings and materials for primary schools | | | ✓ | TT4 |
| Maptionnaire | | | ✓ | TT5 |
| Citizen Science | | | ✓ | TT6 |
| Storymaps | | | ✓ | TT7 |
| Travel guide to climate robust river landscapes | | | ✓ | TT8 |
| Design thinking—Embassy of Water | | | ✓ | TT9 |
| Citizen meetings | | | ✓ | TT10 |
| Landscape Fund | | | ✓ | TT12 |

* is used if the tool appears in more than one classification.

### 4.3. Step 2: Classification

The online workshop was organized for the water professionals to adequately classify the tools and fill in the questionnaire. This was used to exchange knowledge between the WPs coming from different countries, and define what knowledge, transition, and co-creation tools are from the user's perspective. Based on this information, participation tools are used in three interconnected stages within the design process of NbS such as:

#### 4.3.1. Co-Creation Stage

In this stage, co-creation tools are used to support collective creativity specifically the NbS design. Co-creation is used to shift the focus from centralized governance toward a more shared decision-making approach by empowering local citizens and encouraging strong partnerships [47,48].

#### 4.3.2. Knowledge Stage

This stage requires the use of Knowledge Tools (KT) which support the WPs including the stakeholders to observe what is happening in the SES and to construct stories about what could happen. Knowledge tools (KT) support water professionals in the processes of continuous learning and ecological knowledge and understanding.

#### 4.3.3. Transition Stage

Transition Tools (TT) support water professionals in the transition process towards NbS. TT can be policy instruments and measures that facilitate the adaptation of complex systems to changing internal and external circumstances [49].

In this research, a Transition Tool is defined as a tool that supports the initiation and/or process of transition from the existing water management situation to a stable envisaged climate-adapted water management practice—in our case with NbS. This could be the transition from a highly productive grassland with a fixed groundwater level to a 'wetland' with a natural groundwater level. The latter has an increased adaptive capacity to climate change.

After all the tools are collected, selected, and classified by using the first two steps of the stepwise framework, we have three resulting tables where the comparable MC classifications are presented together with the descriptive characteristics such as the tools' descriptions, strengths, and weaknesses, and practical requirements (Table S2).

### 4.4. Step 3: Grading

As mentioned already in Section 3.4, the usability index uses a combination of qualitative and quantitative measures that capture the usability (simplicity/complexity) and the suitability for different applications of these participatory tools that are used by water professionals. The File S2 describes the scoring for each of the sampled participatory tools and offers justification for the score based on the judgment of the authors.

The usability index has a threshold between 5 for the simplest and 18 for the most complex tools according to the criteria presented in Table 3. In our case, the participatory tools used in the Co-Adapt project have usability indexes varying from 5 (TT1) to 15 (KT1). There are no major differences between the usability index for the 3 different tool types. This is evident if we compare the average and standard deviation values for the co-creation (10.3; 2.0), knowledge (10.1; 2.3), and transition (9.5; 2.0) tool groups respectively.

### 4.5. Step 4: Mapping

In total, six different bubble plots are displayed (Figure 4). Colored bubbles represent the individual participatory tools with the radius of the bubble showing the usability index respectively. Numbers indicate the usability index. Two plots are available per tool category (knowledge, co-creation, or transition). The first plot is showing the tools' characteristics linked to the tool type versus the aimed stakeholder type and the second plot displays the purpose of the tool where the objective of the tool is plotted versus the stage of the

NbS project where the tool can be used. While the categories used (tool type, stakeholder type, NbS design process, and objective of the tool) were selected to best capture the overall application and use of a tool, some tool attributes could not be included and are additionally presented in Table S2. Tools' objective only presents the primary purpose for which the tool was designed. Nevertheless, one tool can have more than one objective, and this is also presented in Table S2 and could potentially be another determining factor for tools' selection.

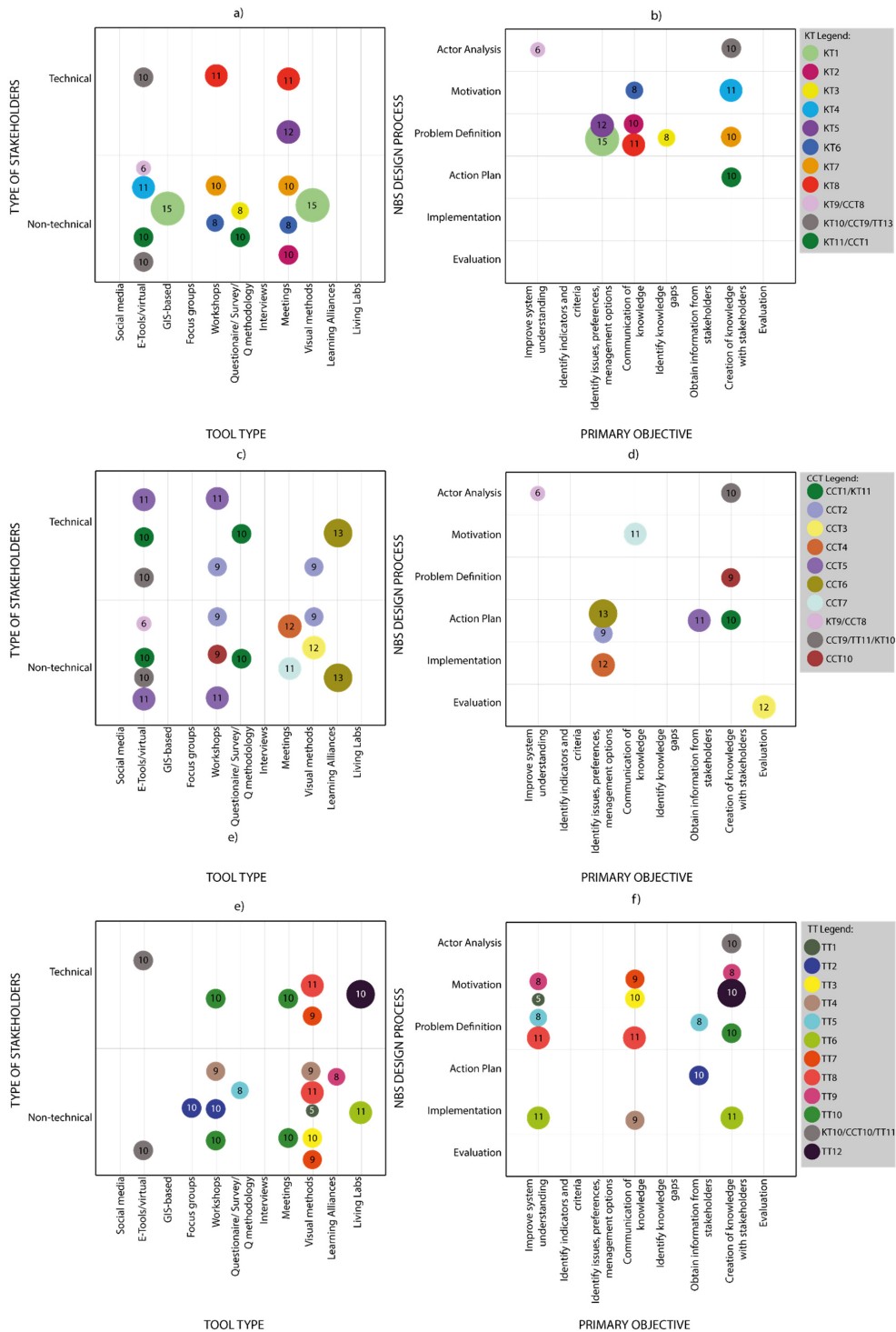

**Figure 4.** Tool mapping: Values are not continuous, the distance from the axes has no effect. Some of the participation tools fall into more than one tool category and are presented in the respective category with the same color (**a**,**b**) co-creation, (**c**,**d**) knowledge, (**e**,**f**) transition tools).

If we look at the first columns for the tool category (Figure 4a) it is evident that social media and interviews were not used in any of the eight catchments. Social media was not used mainly for the reason that, for dissemination purposes, the social media channels of the WPs were used instead of separate social media channels for the particular project in every catchment. Interviews were also not used in the Co-Adapt project for a combination of reasons namely the COVID-19 restrictions on face-to-face visits and preference over group meetings and workshops. For the second column we can notice that in the Co-Adapt, the identification of indicators and criteria is not a preferred objective.

Furthermore, we will explore the differences between the co-creation, knowledge, and transition tools. For the co-creation tools (Figure 4a) we observe that the number of technical and non-technical tools is equally divided. Moreover, e-tools/visual tools and workshops are the preferred tool in the catchment (Figure 4a). As a primary objective (Figure 4b), we observe that tools were used in all design processes. Knowledge tools are used mostly with non-technical stakeholders (Figure 4c). Additionally preferred tool type in this category of tools is again the e-tools/visual tools and meetings (Figure 4d). In Figure 4e, we can detect that knowledge tools were not used in the implementation and evaluation stages of designing NbS. With transition tools, there is a slight preference for tools used for non-technical stakeholders in contrast to technical stakeholders. As the most popular tool type in this category, we have the visual methods (Figure 4e). The only stage where tools were not used is the evaluation stage (Figure 4f).

### 4.6. Tooling Insights and Trends

WPs had to be fairly inventive to co-design with stakeholders in times of the COVID-19 pandemic with so many restrictions taking place, limiting live encounters particularly. This resulted in increased use of e-tools and virtual methods for making the co-creation process possible. Completely contrasting examples in terms of the usability index are the following: 'The walking app' (KT9/CCT8) is a virtual mobile application aimed to guide the stakeholders in the affected area with pre-made walking trails for both children and adults, including geo-tracked information on the trails about specific 'hotspots' about the planned NbS and fun interactive surveys along the way. The fact that this visual tool is both used as a knowledge and co-creation tool and that the usability index is quite low (6) making it part of the 'simple tools' spectrum, makes it additionally attractive for other professionals to use the same tool or even build on it and develop it further based on their local needs. On the other hand, if we look at the 'hydrological modeling tool' (KT1), which is a knowledge tool that has the highest usability index score (15), making it a 'complex' tool to use with stakeholders, WPs might feel more discouraged to use it if their means and time are more limited and choose a simpler tool instead. The way this scoring would help WPs is that it would potentially save them valuable time developing or searching for a suitable tool or suite of tools, in times where time, especially in a project such as the Co-Adapt is quite valuable and scarce. Furthermore, if we compare all tool types, we can notice that not a lot of tools were used for the evaluation stage of the NbS process. Explanation to this could be because most of the projects are busy with implementing the measures and the evaluation stage is yet to come. The same applies also to evaluation but this time was seen as an objective and not a stage in the process.

### 4.7. Case Study Comparison on Participatory Tool Selection: Pre and during Pandemic

Let's zoom in on the Laakbeek case study (Table 1. We can showcase this case study as an example of the use of tools with stakeholders—pre-pandemic. The WPs commenced the project with a preliminary idea of creating a park with swings and playgrounds that will be used most of the year, however in flooding periods it will be repurposed into a buffer zone where water will be stored and slowly discharged to the Laak River. The tooling catalog of the province looked in the following way:

1.     a website of the project where online tools were incorporated such as:

    a.     'views on the climate issues survey' (KT3) in the problem definition stage,

       b.     'online ideation' (CCT1/KT11) in the action plan stage,

2.    live workshops and field trips such as:

       a.     'landscape planning' (CCT2) in the action plan stage,

       b.     'final design presentation' (CCT3) in the evaluation stage, and

       c.     'stakeholder forum' (TT2) in the action plan stage.

3.    'hydrological model' (KT1) of the catchment in the problem definition stage.

By doing this, the WPs managed to educate the population about the climate effects and benefits of using NbS as adaptation measures by showing them the results of the hydrological model and how the catchment can be influenced by the flooding in times of climate change. What we consider especially important with the use of this tool is the information acquired from what the stakeholders essentially consider to be the problem in the catchment, and that differed from the WPs' preliminary design. The 'landscape planning' tool gave the stakeholders a safe and interactive environment where they can express their needs and concerns. This particular case encompassed a safe area where citizens can transit by keeping the catchment as natural-looking as possible. Furthermore, the stakeholders had the opportunity to design the catchment in groups by using an 'imaginary' budget as a monetary restriction that exists in real-life situations. The deigned area where NbS are planned to be implemented is relatively small (1.57 ha), so this could affect the higher level of communication used and the freedom to what can be implemented there as adaptation measures. Another factor could be due to tooling being used in a physical setting, pre-pandemic where it is easier to deliberate and share ideas.

In contrast, the Somerset catchment (Table 1)is an example of a bigger catchment that used tooling during the pandemic. This created a demand for the innovation and development of new online participatory tools. The result was—the development of two new tools using two-way communication and participatory decision making respectively. The walking app (KT9/CCT8) had the main objective to improve the systems' understanding of the stakeholders in relation to their natural surroundings and was used in the actor analysis stage. For this, two-way communication is used in such a way that the stakeholders received detailed videos explaining the planned NbS based on their geo-location accompanied by surveys for the stakeholders to give feedback and ask questions. The other tool is called the 'Adaptation Pathways tool' (CCT9/TT11/KT10) used also in the actor analysis stage. This web-based tool allows stakeholders to vote on adaptive pathways (suite of NbS measures in the catchment) and analyze their effects. Additionally, stakeholders can modify the existing adaptation pathways or make new ones for which other stakeholders could also vote.

Despite the current situation of constant lockdowns and unpredictability of how the future would look, it was quite interesting to see that water professionals still managed to design and facilitate tools to the level of participatory decision making with the limited means and time that was provided. We are confident that the framework, supported by the NbS toolkit will deliver the needed guidance and selection aid for participatory tools and aid in the uptake of designing NbS in a participatory way.

## 5. Conclusions

Designing nature-based solutions with stakeholders is now an emerging practice, which has moved beyond the mere conceptual stage. Practitioners have been developing tools for stakeholder-inclusive nature-based solutions design or have been choosing tools from a quite 'saturated' market of already existing tools [22]. This study developed and tested a stepwise tool selection framework that supports the water professionals toward the stakeholder-inclusive design of NbS. The significance lies in two main reasons. Firstly, as far as we are aware, it is the first to propose a framework for making an informed and systematic selection of participatory tools for the design of NbS. The subsequent visuals can influence prospective users (WPs) for identifying the tool(s) best suited for specific requirements. Secondly, the established set of principles and lessons learned for the use

of participatory tools in brook catchments could be applied for participatory design in different sustainability contexts.

In our study, the pathway toward stakeholder-inclusive nature-based solutions design is through defined three stages, defined by the WPs themselves, with their respective tools. These are operationalized in such a way that: (i) knowledge tools provide stakeholders with new systems' understanding of the environmental challenges in the face of global change and could be applied in the long term beyond the temporal and planning targets of the initial participatory processes, (ii) co-creation tools, can merge local contextual knowledge with scientific knowledge into the design of SES management practices and (ii) transition tools help decision-makers and other stakeholders to translate system understandings into actual decisions.

If following the stepwise framework proposed in this research, we aim to scaffold water professionals to make an informed choice of tools based on tool usability and specific contexts. Additionally, we aim to bridge the gap between the heterogeneity of stakeholders collaborating to design and implement NbS in brook catchments and other SES.

Based on our study findings, we see that among the WPs: (1) knowledge tools are central in the problem definition stage, particularly with non-technical stakeholders; (2) most anticipated co-creation tools are e-Tools/Virtual tools and workshops; (3) transition tools favor visual tools as a way of enabling the transition towards management practices.

We have additionally adapted the simplicity/complexity index for water-energy-food nexus tools from Dargin et al. (2019) [40] to a usability index for participatory tools and tested it on 28 tools used in eight catchments. Subsequently, we mapped the tools against valuable tools' characteristics and their respective usability indexes, to provide the necessary support in choosing the most appropriate tool or suite of tools based on the local contexts and requirements of the specific SES. The resulting visuals can give added value to prospective water professionals by offering a usability spectrum relative to different categories. Having an overview of the tools' relationships and trends can result in strengthening existing tools by using those in the preferred process (knowledge, co-creation, or transition) and stage of the NbS design (Actor analysis; Motivation; Problem Definition; Project Definition; Action Plan; Implementation or Evaluation). The usability index could be a potential requirement for co-designing novel tools for the changing needs and requirements of the SES. We hope that this will initiate a new discussion on the notion of usability among participatory tools.

We advise practitioners to utilize this stepwise framework before designing their participation process with tools. After the selection of tool/s, in theory, this framework can extend, at a later stage, the wide-ranging framework for understanding participatory processes from Hassenforder et al. (2015) [17]. By doing this, we offer practitioners not only a decision-support framework for the selection and analysis but also the effectiveness of participatory tools. Other projects that aim at stakeholder-inclusive NbS design could inventorize their plethora of tools following our stepwise framework and augment the extensive NbS catalog of tools from Voskamp et al. (2021) [22]. Furthermore, enriching this catalog with our usability index and mapping step would give future practitioners a greater market analysis and improved trend insights. Practitioners often require simpler, easy-to-use, and understandable tools that can be readily incorporated into participatory processes [50,51]. This is additionally utilized by giving quantitative associations to the tools with our suggested methodology on the usability index.

The six mapping plots are prototypes for visualizing the capabilities and purposes of tools in terms of their usability. This kind of framework could be conceivably extended as a web-based inventory, with filtering mechanisms to improve the collection, selection, classification, grading, and mapping process of participatory and similar tools.

Ultimately, we agree with Delacámara et al. (2020) [52] that it is up to us and our choices to determine whether social-ecological systems evolve in a resilient and integrated way or not. Whereas our suggested framework offers decision-support on the participatory tools, it does not guarantee openness and transparency towards the stakeholders. The

integration fundamental to the recommended stepwise framework consists in facilitating the above-mentioned interactions of social-ecological systems. If these links are cooperative and transparent, in combination with our suggested stepwise framework, we are poised that relationships between the SES subsystems will be created and strengthened particularly in times of global change where hydro-meteorological hazards are becoming more complex and ambiguous.

**Supplementary Materials:** The following supporting information can be downloaded at: https://www.mdpi.com/article/10.3390/su14095562/s1, Supplementary Materials Tables S1 and S2, Files S1 and S2.

**Author Contributions:** Conceptualization, B.B., A.L., J.H. and S.C.D.; methodology, B.B.; formal analysis, B.B.; investigation, B.B.; writing—original draft preparation, B.B.; writing—review and editing, B.B., A.L., J.H. and S.C.D.; visualization, B.B.; supervision, A.L., J.H. and S.C.D.; funding acquisition, A.L. All authors have read and agreed to the published version of the manuscript.

**Funding:** This research was conducted as part of the Co-Adapt: Climate adaptation through co-creation project, funded by the INTERREG 2 Seas/Mers/Zeeën Cross-border Cooperation Programme 2014–2020, project number 2S06-023.

**Institutional Review Board Statement:** Not applicable.

**Informed Consent Statement:** Not applicable.

**Data Availability Statement:** Not applicable.

**Acknowledgments:** This research would not be possible without the work of the the guidance of Judith Floor and four bachelor students from the Environmental Sciences department at the Open Universiteit, mostly in terms of data collection on transition and knowledge tools through the questionnaires. For the design and collection of the questionnaires used for co-creation tools, we would like to say a big thank you to our project partners from the VLM (Vlaamse Landmaatschappij), the Dutch Province of Noord-Brabant, and Co-Adapt partners. We also thank all project partners in Co-Adapt for participating in the workshops and providing the information for this research. Lastly, we also feel very grateful for the support from Dave Huitema especially on the social science aspects of this paper and its conceptualization.

**Conflicts of Interest:** The authors declare no conflict of interest.

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
