# Peer review of "Participatory Design of Nature-Based Solutions: Usability of Tools for Water Professionals"

_sustainability, doi:10.3390/su14095562_

Round 1

Reviewer 1 Report

Dear Authors, I am sorry but I must state that I have not found anything new in your manuscript in the field of science. The methods you use have been tested before and do not represent progress, and the results obtained are expected.
I can conclude that the manuscript is below the level for publication in a respected scientific journal.
Keep researching and working, I'm sure your next manuscript will be much better.

Best regards.

Author Response

Dear reviewer,

Firstly I want to thank you for taking your time to read the manuscript and give us feedback. Secondly, we find it quite important to know a bit more into detail how it can be improved. You mentioned that the methods have been used before, hence there is not enough progress. I would kindly argue with this statement because we indeed use a scoring framework from another study, however the context and outcomes are different and a lot of studies use existing methodology and build upon it. If this is maybe not evident in the current version of the manuscript please do let us know. According to us, the significance lies in two reasons. As far as we are aware, it is the first to propose a framework for making an informed and systematic selection of participatory tools for the design of NbS and establish a set of principles and lessons learned for the use of participatory tools in brook catchments that could be applied for participatory design in different sustainability contexts. I am attaching another version of the manuscript where we also tried implementing the comments from the other reviewers. In this version, results and discission are merged together because we agree with the reviewers that this brings an added value of presenting them in one chapter. The conclusion chapter is also sharpened in a way that will showcase better what we did in contrast to other research and how to move forward. The figures and tables are improved and the appendixes are made a little bit shorter. This is what we could do in the time frame given from the editor (one week) which was quite challenging considering it is a 'major review' so I hope this will possibly be taken into account.

Looking forward to receving further valuable feedback!

Reviewer 2 Report

 Participatory design of Nature-based solutions: Usability of Tools for Water Professionals

The research paper discusses an interesting point, and the research gap is presented in the abstract. The authors have presented their work aligned on a stepwise framework to use the participatory tools in effectively. However, there are few slips in research, and I would like to state them herein.

Abstract –What are your findings and how you market your research work to the readers? The research findings are not really presented in the abstract. This is a major loss of your paper.

Introduction – “For this purpose, we propose a stepwise framework to categorize different tools in the participation processes and develop a methodology for calculating the usability index per tool.” – Can the authors showcase the novelty in this approach?

Good count of references is in the introduction.

Locations given in Figure 2 not clear enough. Can the authors, showcase these locations (may be a table)?

Figure 3 is very unclear. Can the authors improve this figure as this is one your major results.

This is a very interesting study.

However, I am struggling to find the real output of the carried research work. The authors have separately presented the results and discussion. Yeah, that is fine.

I personally like results and discussions to be in one section rather than 2. However, it is upto the authors to decide it. There is an advantage when you have a combined section; you can present your results and discuss the merit of them as well as the limitations of them.

However, what are your real conclusions from the research. Even though, the manuscript was prepared well, the real outcome of your research is missing. Therefore, I have to move with major revisions for your paper. Please revise it and showcase your conclusions.

One major issue in your paper which I have noticed is the presentation. Can the authors have a deep look at it and restructure the results section (or with discussion). Can you also highlight the facts that you have addressed under the showcased gap?

Reviewer 3 Report

Several major issues:
1. The describe about methodology should be improved. Step 1 and Step 2 should be combined.. The so-called "Classification" is essentially about when to use the tools. In Section 3. Results, Table 3 should be bettwe re-organized. For instance, list the tools in three columns: Tool type (or Tool Code), Tool title and  Application stag. 
2. How to map the assessment results in Step 3 should be prensented in more details.
3. There are too many appendices. First of all, there are some unnecessary deuplicates, such as forms Knowledge Tool 2, Knowledge Tool 3, Co-Creation Tool 2. In fact, the forms for Knowledge Tool , Co-Creation Tool and  Transition Tool are very similar, so it is better to make one form for all. Secondly, I suggust reorganizing the materials and puting some of them in the main text, especially about Grading, which should be a main content of the framework for the use of participatory tools, rather than in the form of an appendix.
4. "BOX 1: CASE STUDY COMPARISON ON PARTICIPATORY TOOL SELECTION 396 PRE AND DURING PANDEMIC" is actually not a box. It should be a normal section, like 4.5. The major problem is, the case study is not well describde, especially, how the tools are applied in each stage, such as what kind of hydrological model. I strongly suggest presenting more details about the case study.

5. I suggest differentiating discussions from conclusions.

Some details:
1. The contents of Table 1 is not well organized, which make the meaning of Table 1 not quite clear.  And, Section 3 should talk about the methodology generally, so  the cathment names should not be listed in Table 1.
2. The meaning of QT, KT, TT , and CC etc. should be given beneath Table 1.
3. It is not logical presenting 3.2.1 without 3.2.2.
4. Fonts in Figure 3 are too small.
5. There are duplicated "Section 4.5".  "4.5 Tooling insights and trends" is not about results. It chould be considered as a part of Discussions.
6. Appendix A is missing.

Round 2

Reviewer 2 Report

Revisions are acknowledged. 

Reviewer 3 Report

I'm mostly satifisfied with the revision.